# Elemental and K-Ar Isotopic Signatures of Glauconite/Celadonite Pellets from a Metallic Deposit of Missouri: Genetic Implications for the Local Deposits

**Norbert Clauer [1,*], I. Tonguç Uysal [2,3] and Amélie Aubert [1]**

[1]  Institut des Sciences de la Terre et de l'Environnement de Strasbourg, Université de Strasbourg (UdS/CNRS), 67084 Strasbourg, France
[2]  Department of Geological Engineering, Istanbul University–Cerrahpaşa, Istanbul 34098, Turkey
[3]  School of Earth and Environmental Sciences, The University of Queensland, Brisbane 4072, Australia
[*]  Correspondence: nclauer@unistra.fr

**Abstract:** In the course of attempting to date the host rocks of Viburnum metal deposits from the US state of Missouri, the purpose was here a detailed examination and contribution of the constitutive minerals of glauconite-rich pellets to the isotopic dating of these deposits. The glauconite pellets of Cambrian sediments hosting metal concentrates were dated here by the K-Ar method to complement earlier published Rb-Sr data. The study confirmed that the preparation and purification step of such glauconite pellets is especially critical with the need for a specific cleaning step to not only remove the detrital counterparts but also all Sr-rich components occurring as accessory minerals such as the carbonates, sulfates and oxides that apparently "contaminated" the Rb-Sr results. The K-Ar data and the previously released Rb-Sr results obtained on strictly the same glauconite-rich separates outline clear age discrepancies that can be summarized by higher, "older" K-Ar age data at about 440, 415 and 390 Ma, and lower, "younger" Rb-Sr data at about 400 and 370 Ma. The glauconite separates of most samples being apparently not contaminated by various detrital K-rich crystals, the two dating methods should have been affected similarly. The analytical dispersion seems, then, to result from a diagenetic event that affected the Rb-Sr system more than the K-Ar system by a plausible addition/subtraction of one or several Sr-rich and Rb-poor and, therefore, K-poor minerals. In turn, the studied pellets were apparently impregnated after deposition by flowing metal-rich fluids in a low-temperature environment not affected by a significant thermal impact. The Bonneterre Formation acted apparently as a regional drain for metal-rich fluids that percolated throughout the region at a probable burial depth of less than 2000 m.

**Keywords:** glauconite pellets; major and trace elemental compositions; K-Ar isotopic dating; Cambrian Bonneterre Formation; Mississippi Valley Type deposits; Missouri U





## 1. Introduction

Often identified as "green pellets" of sediments, authigenic glauconitic minerals, when purified, provided numerous Rb-Sr and/or K-Ar isotopic ages over time that contributed absolute ages to the geological time scale. That important contribution of glauconite fractions for the calibration of the geological time scale has resulted in their inclusion in many studies (e.g., [1–3] and many more). Indeed, since Cormier's [4] pioneering Rb-Sr age dating of glauconite-type materials, many Rb-Sr and/or K-Ar analytical studies of this mineral phase have been published (e.g., [5–8]). The $^{40}$Ar/$^{39}$Ar method was even applied on purified glauconite and considered successful [9]. However, being based on nuclear irradiation and subordinate recoil, the application of this latter method is questionable, at least when applied to K-type clay glauconite, whose granular composition does probably not control and therefore does not minimize the recoil effect. Moreover, the nanometer size of the constituent crystals of the grains does not favor closed-system behavior during nuclear

irradiation [10,11]. Even the ages of glauconite separates determined by the classical K-Ar method can be questionable (e.g., [12]. In fact, the concerns raised in these evaluations focus mostly on preparation methods. For instance, Derkowski et al. [13] estimated the dissolution rates of the "polluting" detrital minerals of glauconite pellets by various interactions with solutions of extremely low pH. They found that the K-Ar ages of the pellets increased with increasing reaction time, which indirectly confirmed the partial dissolution of late-digenetic authigenic non-glauconitic components. Odin and Hunziker [14] detailed preparation steps of green aggregates by testing a procedure to remove the foreign particles of the glauconite pellets, authigenic or detrital, based on the combined use of dilute acetic acid and ultrasonic shaking. Keppens [15] used dilute hydrochloric acid combined with ultrasonic cleaning, whereas other authors preferred an ammonium-acetate leaching without shacking or vibrating impacts (e.g., [6,16,17]. In this context, Clauer et al. [18] examined the impacts of various published preparation methods on various clay-type size separates. That study demonstrated the techniques that are inappropriate for glauconite purification, but these considerations were considered provocative at the time the publication was released, even though it was not intended to rank individuals but the applied methods. The reason for age discrepancies can be attributed to choices of the separation techniques of clay-type materials, especially of glauconitic pellets because of their granular texture that can favor hosting diagenetic minerals during the post-depositional evolution of the pellets. For instance, and based on the described preparation methods, some of the separated material may have been disaggregated further, dissolved or even lost during the successive steps of the experiment. Recently, Clauer et al. [19] addressed the impact of ultrasonic shaking on glauconite separates in addition to gentle leaching by various diluted reagents. They compared the obtained K-Ar and Rb-Sr ages and found that age differences may be due to the occurrence of Sr-rich soluble minerals that might affect the Rb-Sr age values but not the K-Ar corresponding data. This analytical aspect was addressed again here by considering the K-Ar and Rb-Sr data of strictly the same glauconite pellets analyzed in two different laboratories with the Rb-Sr data published long ago by Posey et al. [20].

The Viburnum metal deposits have been dated, both directly and indirectly ([20–23]). The purpose then is not to add to the available library of data and interpretations. The aim is rather to complement an earlier Rb-Sr study by examining the chemical compositions and the K-Ar age data of the same separates analyzed before by Posey et al. [20] for a more complete evaluation of the isotopic ages of minerals involved in the metal genesis. Whilst the Rb-Sr data of Posey et al. [20] might appear less reliable than can now be achieved by the present-day mass spectrometers, especially with regard to the accuracy of generated $^{87}Sr/^{86}Sr$ ratio, it also cannot be denied that the Rb-Sr isotopic method was already, at the time of their publication, of high quality considering the intrinsic characteristics of the mass spectrometers at the time.

## 2. Geological and Regional Mineralization Context

Weak remnant magnetism analyses of ore and barren host rocks of southern Missouri by Beales at al. [24,25] and Wu and Beales [26] defined late Carboniferous to early Permian paleo-pole positions that these authors interpreted as an index for the timing of some regional mineralization. McCabe et al. [27] also reported late Paleozoic pole positions in the Bonneterre Formation of southeastern Missouri, while York et al. [22] obtained an age of 549 ± 20 Ma by $^{39}Ar/^{40}Ar$ dating of pyrite, also in southeastern Missouri. Despite scattered data and very low K and radiogenic $^{40}Ar$, the authors suggested a Cambrian age for the southeastern Missouri mineralization concentrates, while Kish and Stein [21] published earlier a younger Rb-Sr isochron age 358 ± 6 Ma for glauconite separates from the Magmont Mine area, located as well in southeastern Missouri. These glauconite separates were considered by the authors to have interacted with flowing fluids during a late Devonian-Early Carboniferous mineralizing event.

Brannon et al. [28] considered genetic models for the tenuous precipitation of Mississippi Valley-Type (labeled MVT hereafter) ore deposits due to the lack of direct isotopic

dating of the potential events associated with the ore deposition/concentration. The same authors argued that the available ages support a model that attributes the formation of several, but not all, North American MVT deposits to long-range migrations of basin brines in response to the Alleghenian-Ouachita orogeny (325–260 Ma). Although agreeing that the formation of MVT deposits is controlled by basin brines Nakai et al. [23] raised concerns about the exact origin of the North American MVT, partly because of the scarcity of reliable geochronological data. The Rb-Sr-dated sphalerite from Immel mine in East Tennessee at 347 ± 20 Ma is analytically consistent with the Rb-Sr age of 377 ± 29 Ma of sphalerite deposits from nearby Coy mine but inconsistent with models that ascribe the genesis of this mineral to secondary effects of the late Paleozoic Alleghenian orogeny (325–260 Ma). In this genetic context of the MVT deposits in southeastern Missouri, the so-called Central Dome contains pyrite-chalcopyrite deposits that yield $\delta^{34}$S values suggesting temperatures of 275 ± 50 °C (Shelton et al. [29])). In a large review of the MVT deposits throughout geological time, Leach et al. [30] provided an extended summary of the available age data from deposits of the Ozark Mountains between 380 and 350 Ma for Rb-Sr data on glauconite grains [31] and at 392 + 21 Ma for galena [32]. Older ages by $^{40}$Ar/$^{39}$Ar determinations on pyrite at 549 ± 20 Ma [22], as well as a wide range of data from 489 ± 8 to 297 ± 7 Ma on clay pods, were considered to be associated with mineralizing fluids [33]. In summary, all regional geochemical, isotopic and genetic studies on the mineralization episodes of the southeastern Missouri and of analogous districts in mid-Pennsylvanian (late Carboniferous) rocks [34–37] agree with an overall southeastern Missouri MVT mineralization event during the late Carboniferous to early Permian.

The first attempt to date the ore concentrates of southeast Missouri was by York et al. [22], who applied the $^{40}$Ar/$^{39}$Ar method on pyrite. The obtained age of 549 ± 20 Ma is questionable as it predates the host-rock time of deposition. Among the other attempts to date the southeastern Missouri ores with the Rb-Sr method is that of Lange et al. [32], who used fluid inclusions in galena to obtain a date of 392 ± 11 Ma that was contested by Ruiz et al. [38] who retracted their arguments in a further discussion [39]. The Rb-Sr dating of glaucony separates from Bonneterre Formation yielded ages ranging between about 350 and 400 Ma [20,31,40,41]. These results suggest that the ore-forming fluids in the Viburnum Trend varied substantially in their isotopic results, whatever the method. Widely scattered K-Ar data of illite samples from the same Viburnum Trend range from 489 + 8 to 297 ± 7 Ma, which was considered a minimum date for the mineralization event [32]. The youngest of these age dates appears to be consistent with independent regional geological and geochemical evidence [34–36] suggesting that the ores in southeastern Missouri relate to a widespread MVT mineralization younger than the Mid-Pennsylvanian rocks of about 300 Ma. In summary, the available isotopic data are scattered over a large time interval, which could be due to the wide variety of dated materials, for which the isotopic records were not perfectly constrained by appropriate sample preparation/identification steps.

## 3. Description of the Samples and of the Analytical Procedure

The analyzed glauconite-rich samples were collected from cores drilled into the Viburnum Trend in the state of Missouri (Figure 1). Most of them belong to the dolomitic Cambrian Bonneterre Formation, while a few correspond to the transition zone with the underneath Lamotte Formation. The samples were hand crushed in a ceramic jaw crusher, and the powders were shaken during 20 min in a rotating tap and sieved at an 80–120-mesh size. This size fraction was then processed through a Frantz magnetic separator to isolate the magnetic glauconite pellets from carbonate and quartz agglomerates and, occasionally, from more magnetic hematite grains. To be mentioned here is the fact that depending on the size of the pellets or grains, the separation sometimes had to be improved by hand picking. Then, the samples were washed with de-ionized water and rinsed further with dilute 0.1 N HCl to remove the calcite and hematite particles adhering at or incorporated in the pellets. Then, the samples were air-dried and processed for the elemental and isotopic analyses; some being also X-rayed. A few observations of thin sections showed some

remaining carbonate inclusions within the glauconite grains. All these steps are from the procedure of Posey et al. [20].

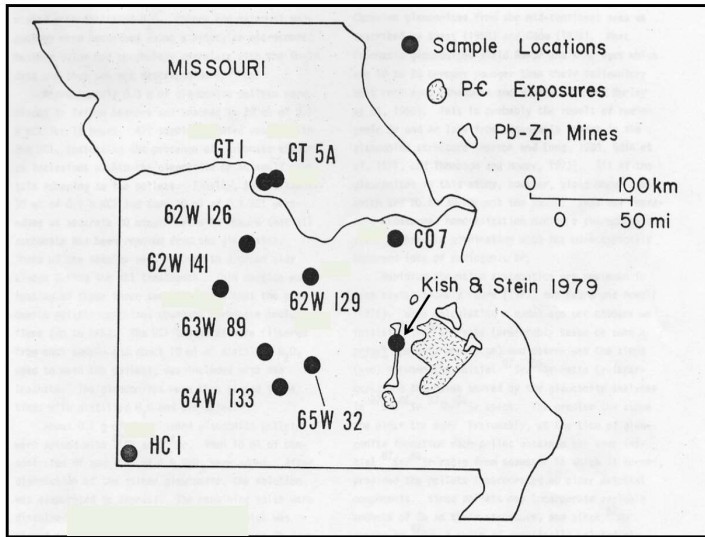

**Figure 1.** Location of the drill holes from which the glauconite-rich samples were collected in the Viburnum Trend of Missouri State (map of Posey et al. [20], slightly modified).

Some of the powder separates were also analyzed here by X-ray diffraction (XRD) for a complementary identification of the pellet components. The major elements were quantified with an induced coupled plasma-atomic emission spectrometer (ICP-AES), while an induced coupled plasma-mass spectrometer (ICP-MS) was used to analyze the trace elements, both techniques following the procedure of Samuel et al. [42]. The analytical accuracy of the results was permanently controlled by the interpolation of the samples with known international standards, such as BE-N and GL-O, which data gave an internal precision of $\pm 2.5\%$ for the major elements and of $\pm 5\%$ for the trace elements. The Ar extractions were obtained in a glass extraction line connected to a static gas mass spectrometer. The samples were first heated moderately under vacuum at 80 °C for at least 12 h before analysis to reduce the amount of atmospheric Ar that could have adhered to the pellets during preparation and handling. The accuracy of the K-Ar method was checked weekly by measuring the glauconite standard GL-O that averaged $24.59 \pm 0.17$ (2σ) $\times 10^{-6}$ cm$^3$/g (STP) radiogenic $^{40}$Ar for 5 independent measurements at the time of the study, to be compared to the recommended standard value of $24.85 \pm 0.48$ (2σ) $\times 10^{-6}$ cm$^3$/g [43]. The procedure also included periodic determinations of atmospheric $^{40}$Ar/$^{36}$Ar loads that averaged $298.7 \pm 1.2$ (2σ) to be compared to the recommended value of $298.6 \pm 0.4$ (2σ; [44]). The blanks of the coupled extraction line and mass spectrometer were controlled once a week. Their contents never exceeded $1 \times 10^{-8}$ radiogenic $^{40}$Ar, being more often below $1 \times 10^{-9}$, which means that the equipment did not add measurable residual $^{40}$Ar to that of the samples. The K-Ar ages were calculated using the recommended decay constants [45] with their individual analytical uncertainties. Further details of the analytical procedure are available in Bonhomme et al. [46], while the earlier used Rb-Sr procedure was detailed in Posey et al.'s [20] publication.

## 4. Results

Some of the glauconite separates dated previously by the Rb-Sr method [20] were dated again here by the K-Ar method and analyzed again by XRD, as recent studies have shown that the preparation step of isotopically dating such complex material may be determining (e.g., [19]) when separates contain glauconite but also celadonite. The latter can be considered a crystallographic variation of glauconite with Al in the tetrahedral position together with Si and Mg in the octahedral sites instead of Al and Fe and Na in

the interlayers with K [47,48] or considered to have a different structural and chemical organization (e.g., [49–51]) Note, this does not mean that these two minerals are of the same generation or origin (e.g., [52–54])Four of the five separates also contain dolomite, which is a major component of the dolomitic Bonneterre Formation, while another separate contains calcite instead. Pyrite and/or hematite were also identified in five of the separates, together with gypsum and halite. Halite and F-apatite were found in two of the analyzed samples with a small occurrence of quartz in the later (Table 1). This quartz occurrence and probably that of F-apatite suggest a potential occurrence of some detrital minerals. Additionally, the mixed occurrence of glauconite and celadonite in the pellets suggests two independent crystallization episodes.

**Table 1.** XRD data of five of the glauconite-rich pellets studied here.

| Sample IDs | Depth (Feet) | Sample Numbers | Glauconite | Celadonite | Dolomite | Calcite | Hematite | Gypsum | Pyrite | F-Apatite | Halite |
|---|---|---|---|---|---|---|---|---|---|---|---|
| HC 1 | 1776 | 1 | x | x | x | | | x | x | | |
| CO 10 | 1538 | 8 | x | x | | x | x | | | | |
| CO 10 | 1625 | 9 | x | x | x | x | x | x | | | |
| GT 1 | 2095 | 10 | x | x | x | | | x | x | x | |
| GT 5A | 2162 | 13 | x | x | x | | | x | | | x |

IDs stands for identities.

### 4.1. The Major-Elemental Compositions

The total contents of the major elements show a loss of ignition range from 98.1 to 101.0% (Table 2). This overall variation of less than the 2.5% analytical uncertainty consolidates the reliability of the analytical database. In the detail, the elemental contents of only two samples are outside the range of the other fractions. The CaO contents of sample 64CW-133 buried at 1698 feet and of sample 65CW-32 buried at 1840 feet are abnormally high relative to the other analyzed samples. They also yield abnormally low $SiO_2$, $Fe_2O_3$ and $K_2O$ contents and abnormally high LOI contents. Combined, these results confirm the occurrence of calcite crystals trapped within the glauconite pellets.

**Table 2.** Major elemental compositions of the glauconite-rich pellets studied here.

| Sample IDs | Depth (Feet) | Sample Numbers | $SiO_2$ (%) | $Al_2O_3$ (%) | $MgO$ (%) | $CaO$ (%) | $Fe_2O_3$ (%) | $Mn_3O_4$ (%) | $TiO_2$ (%) | $P_2O_5$ (%) | $Na_2O$ (%) | $K_2O$ (%) | LOI (%) | Total (%) |
|---|---|---|---|---|---|---|---|---|---|---|---|---|---|---|
| HC 1 | 1756 | 1 | 51.7 | 12.2 | 6.99 | 1.0 | 9.6 | 0.026 | 0.06 | 0.18 | 0.07 | 7.95 | 8.32 | 98.10 |
| 64W 133 | 1480 | 2 | 49.4 | 7.8 | 3.69 | 0.8 | 21.3 | 0.039 | 0.04 | 0.19 | 0.06 | 7.80 | 7.40 | 98.50 |
| | 1536 | 3 | 47.3 | 8.1 | 3.96 | 3.6 | 20.1 | 0.022 | 0.03 | 0.15 | 0.05 | 7.57 | 8.95 | 99.83 |
| (us) | 1698 | 4 | 32.0 | 6.1 | 9.10 | 12.0 | 13.1 | 0.144 | 0.02 | 0.23 | 0.10 | 4.78 | 23.33 | 100.90 |
| 65W 32 | 1840 | 5 | 21.1 | 3.6 | 12.1 | 17.2 | 11.3 | 0.211 | bdl | 0.20 | 0.09 | 3.47 | 30.02 | 99.29 |
| 62W 141 | 1069 | 6 | 49.6 | 7.7 | 4.77 | 3.0 | 18.2 | 0.014 | bdl | 0.62 | 0.05 | 7.17 | 8.49 | 99.61 |
| CO 10 | 1538 | 7 | 45.4 | 6.5 | 3.16 | 2.2 | 25.6 | 0.060 | bdl | 0.14 | 0.19 | 7.53 | 7.47 | 98.25 |
| | 1625 | 8 | 45.6 | 8.6 | 4.08 | 2.5 | 20.2 | 0.036 | 0.03 | 0.25 | 0.13 | 7.85 | 10.02 | 99.30 |
| GT 1 | 2095 | 9 | 43.8 | 7.3 | 4.34 | 2.9 | 23.4 | 0.071 | bdl | 0.51 | 0.17 | 7.20 | 9.10 | 98.79 |
| (us) | | 9a | 47.7 | 7.6 | 4.22 | 2.6 | 20.9 | 0.028 | bdl | 0.44 | 0.18 | 7.30 | 10.03 | 101.00 |
| CO 7 | 1766 | 10 | 46.0 | 7.6 | 3.33 | 3.4 | 22.3 | 0.021 | bdl | bdl | 0.14 | 7.79 | 9.08 | 99.66 |
| GT 5A | 2162 | 11 | 49.0 | 9.2 | 4.62 | 0.3 | 19.6 | bdl | bdl | 0.12 | 0.47 | 7.96 | 9.34 | 100.61 |
| 62W 126 | 1285 | 12 | 46.4 | 5.8 | 4.74 | 3.9 | 19.8 | 0.035 | bdl | 1.15 | 0.05 | 7.49 | 8.98 | 98.35 |
| | 1400 | 13 | 43.5 | 8.1 | 4.40 | 5.2 | 16.9 | 0.019 | 0.03 | 0.10 | 0.06 | 7.00 | 13.13 | 98.44 |
| 63W 89 | 1295 | 14 | 48.5 | 8.0 | 3.32 | 1.0 | 21.1 | 0.044 | 0.06 | 0.13 | 0.05 | 7.71 | 8.79 | 98.70 |

IDs stands for identities and bdl for below detection limit.

The $P_2O_5$ contents of the fractions 62CW-141 buried at 1069 feet, GT-1 buried at 2095 feet with some F-apatite are high and the ultrasonically equivalent of the sample 62W-126 buried at 1285 feet indicate that except for the fraction subjected to a specific ultrasonic treatment, the P-rich glauconite pellets could have been contaminated by P-enriched fluids that favored the local precipitation of specific P-minerals.

### 4.2. The Trace-Elemental Composition

Among the analyzed metallic trace elements, a few show abnormally high contents in some of the samples: V in the samples 65W-32, CO-10 and GT-1 with and without ultrasonic treatment, Ni in the fractions 64CW-133 at 1460 and 1680 feet depth, 62W-141, CO-10 and the ultrasonically treated GT-1, Cr in 64CW-133 at 1680 depth and CO-10, as well as Zr in the fractions GT-1 with and without ultrasonic treatment and GT-5A at 2118 feet depth (Table 3). Abnormally high Ba and Co concentrations were measured in CO-10, and Cu was also high in the ultrasonic GT-1 separate (Table 3).

**Table 3.** Trace elemental contents of the studied glauconite-rich pellets.

| Sample IDs | Sample Numbers | Rb (µg/g) | Sr (µg/g) | Ba (µg/g) | V (µg/g) | Ni (µg/g) | Co (µg/g) | Cr (µg/g) | Zn (µg/g) | Cu (µg/g) | Sc (µg/g) | Y (µg/g) | Zr (µg/g) |
|---|---|---|---|---|---|---|---|---|---|---|---|---|---|
| 64CW 133 | 2 | 258 | 25.2 | 22.0 | 76.7 | 1806 | 26.6 | 53.9 | 25.9 | 22.1 | 3.9 | 7.3 | 65.3 |
| | 3 | 254 | 31.9 | 36.1 | 51.6 | 60.2 | 39.6 | 56.6 | 36.8 | 12.9 | 7.0 | 9.9 | 57.8 |
| | 4 | 271 | 23.7 | 13.8 | 71.4 | 203 | 20.8 | 104 | 19.5 | 13.1 | 2.0 | 7.6 | 97.4 |
| 65W 32 | 5 | 253 | 32.1 | 32.7 | 108 | 31.9 | 20.6 | 76.3 | 26.8 | 27.1 | 4.6 | 19.4 | 82.2 |
| 62W 141 | 6 | 246 | 34.6 | 32.2 | 61.0 | 638 | 21.3 | 51.6 | 23.2 | 11.3 | 2.2 | 49.0 | 38.2 |
| CO 10 | 7 | 269 | 31.2 | 716 | 125 | 3164 | 91.2 | 6239 | 37.9 | 43.2 | 4.2 | 3.4 | 44.2 |
| | 8 | 281 | 32.5 | 62.0 | 53.3 | 54.8 | 55.0 | 58.5 | 32.8 | 37.0 | 6.2 | 4.8 | 77.0 |
| GT 1 | 9 | 243 | 38.2 | 64.2 | 104 | 30.9 | 14.5 | 60.9 | 31.1 | 19.4 | 7.3 | 24.1 | 132 |
| (us) | 9a | 281 | 38.3 | 77.7 | 107 | 2976 | 28.8 | 6506 | 51.4 | 106 | 7.2 | 23.9 | 135 |
| CO 7 | 10 | 251 | 38.0 | 47.7 | 58.1 | 24.5 | 22.4 | 41.8 | 32.4 | 12.6 | 3.9 | 4.4 | 88.0 |
| | 13 | 243 | 11.5 | 76.3 | 66.2 | 39.4 | 15.2 | 50.6 | 30.4 | 21.5 | 5.2 | 3.0 | 111 |
| 62W 126 | 12 | 262 | 64.5 | 18.3 | 28.3 | 75.1 | 35.9 | 50.3 | 41.2 | 18.3 | 1.7 | 78.7 | 44.2 |

IDs stands for identities.

The CO-10 and the ultrasonically prepared GT-1 pellets contain a combination of at least five metallic elements at high concentrations that suggest a nearby metallic deposit. On the other hand, one further sample also contains at least five metallic trace elements after ultrasonic treatment, which attests to an impact of the ultrasonic shacking on the pellets by increasing the amount of metal-rich grains.

### 4.3. The K-Ar Data

The amounts of radiogenic $^{40}$Ar measured in the analyzed gas of the glauconite-rich separates are systematically between 95.7 and 98.0%, which indicates very consistently positive analytical conditions, especially an efficient purification of the radiogenic $^{40}$Ar, except for the 64CW-133 separate that was further treated by ultrasonic vibrations; the shaking treatment inducing a 23.8% lowering of the radiogenic $^{40}$Ar content. The $K_2O$ contents of the glauconite pellets range quite narrowly between 7.00 and 7.95%, except for the separate subjected to the supplementary ultrasonic treatment for which the $K_2O$ content decreased by 37%. The obtained K-Ar ages range quite widely from 389.9 ± 8.3 Ma to 460.0 ± 9.9 Ma (Table 4).

Harper [55] illustrates that the correlations between the radiogenic $^{40}$Ar and $K_2O$ contents of the analyzed separates are often used in K-Ar dating studies as a control for analytical K-Ar homogeneity/heterogeneity. When the whole rocks or the size separates yield an isotopic equilibrium between the two K and radiogenic $^{40}$Ar, the data points plot on a single array that intersects the origin of both ordinates, indicating in turn a closed and homogeneous behavior. When the separates contain detrital minerals, the data points plot along one or several arrays that intersect the radiogenic $^{40}$Ar ordinate above the intersection with the second ordinate. In contrast, when the analyzed samples here were partly impacted by post-depositional diagenetic events, the obtained array or arrays cut the K ordinate first (Figure 2). Here, the data points did not fit a single straight line but are slightly dispersed in a fairly narrow area and apparently organize into four arrays. Three of the obtained lines fit the intersection of both ordinates at the same time, which suggests an isotopic equilibrium of the separates but also an isotopic heterogeneity for the remaining samples. The upper line through three of the data points does not intersect the common intersection, suggesting some detrital supply, which is consistent with the detection of quartz occurrence in one sample by XRD analysis. Below that mixing line, 2 further lines of, respectively, 5 and

4 data points yield age data of 441.7 ± 6.8 and 416.4 ± 5.7 Ma, while a lower array through the two last data points provides a further date at about 389 Ma. In summary, the K-Ar ages of the glauconite-rich pellets from eastern Missouri MVT deposits that were analyzed here decrease from about 440 to about 390 Ma. These values are significantly higher, "older" if of stratigraphic value, than the Rb-Sr values published earlier that range between about 385 Ma and 368 Ma.

**Table 4.** K-Ar data of the studied glauconite-rich pellets.

| Sample IDs | Sample Numbers | $K_2O$ (%) | $^{40}Ar$ * $(10^{-4}$ cc/g) | $^{40}Ar$ * (%) | Age (Ma) (+/−2σ) |
|---|---|---|---|---|---|
| HC-1 | 1 | 7.95 | 1.11 | 97.09 | 389.9 (8.3) |
| 64W-133 | 2 | 7.80 | 1.31 | 95.72 | 458.2 (9.9) |
| | 3 | 7.57 | 1.16 | 97.81 | 423.5 (9.0) |
| (us) | 3a | 4.77 | 0.77 | 76.55 | 439.7 (14.6) |
| 62W-141 | 7 | 7.22 | 1.08 | 98.02 | 412.9 (8.8) |
| (us) | 7a | 7.17 | 1.15 | 97.80 | 438.5 (9.3) |
| CO-10 | 8 | 7.54 | 1.27 | 97.25 | 460.0 (9.9) |
| | 9 | 7.85 | 1.19 | 95.44 | 416.9 (9.1) |
| GT-1 | 10 | 7.19 | 1.18 | 97.53 | 448.8 (9.6) |
| (us) | 11 | 7.30 | 1.17 | 96.52 | 437.8 (9.4) |
| CO-7 | 12 | 7.79 | 1.24 | 97.44 | 435.2 (9.3) |
| GT-5A | 13 | 7.96 | 1.12 | 97.09 | 389.9 (8.3) |
| | 15 | 7.00 | 1.04 | 96.29 | 412.2 (8.9) |
| 63W-89 | 14 | 7.71 | 1.24 | 92.03 | 442.1 (9.9) |
| 64W-32 | 5 | 3.47 | 0.60 | 60.04 | 468.3 (16.4) |

IDs stands for identities and (*) for radiogenic.

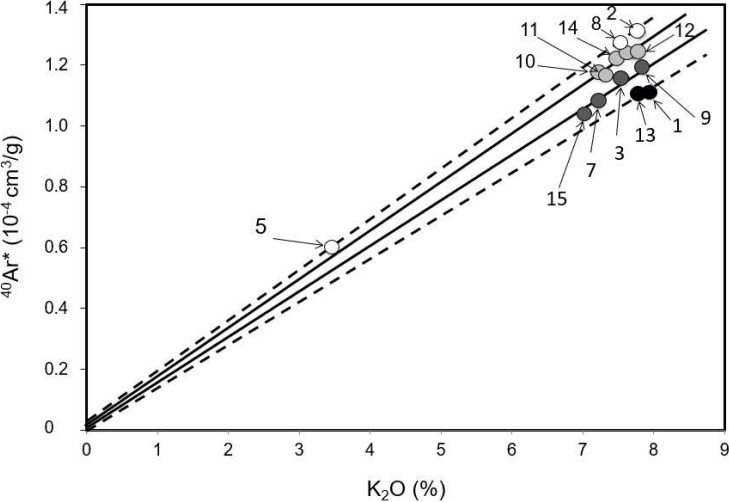

**Figure 2.** Harper [55] display of the K-Ar data from untreated glauconite-rich pellets.

As it has been shown that the preparation step of glauconite material may have a determining impact on the analytical results, three glauconite separates were subjected to a 30-mn ultrasonic treatment in dilute 1 N acetic acid, and their K-Ar ages were determined again (Table 3). In the case of the separates 64W-133 buried at 1536 feet and 62W-141 buried at 1069 feet depth, the K-Ar data have a tendency to increase, within analytical uncertainty for the former and slightly beyond for the second. In the case of the third shaking experiment, the K-Ar result of sample GT-1 decreases but within analytical uncertainty. In sum, the shaking step variably impacts the glauconite-rich separates and again raises the meaning of its impact when applied with dilute acid, either HCl or HAc.

## 5. Discussion

The elemental compositions raise, together with the K-Ar isotopic data, a concern about the observed scatter, most probably due to the heterogeneous mineral compositions of the glauconite-rich agglomerates that reacted differently for the K-Ar and Rb-Sr methods because of the grainy composition of the glauconite-rich agglomerates and, therefore, their mineral and chemical homogeneity. Also of concern amongst the above results is the noticeable difference in the ultrasonic fractions compared with the same separates subjected to the usual preparation. This scatter confirms again that analytical problems appear to be due to an incomplete sample preparation and that this remains a critical step, probably more important than is often assumed [19].

### 5.1. Interpretation of the K-Ar Data

The K-Ar ages of the glaucony-rich separates do not appear to be just dispersed heterogeneously; they rather appear to be organized into successive steps at 441.7 ± 6.8, 416.4 ± 5.7 Ma and about 389 Ma (Table 4). A spontaneous interpretation of such a scatter, whatever the used isotopic method, would be a decreasing occurrence of detrital K-carrying mica-type clayed minerals. However, larger scatters are expected in this case with more dispersed data, such as those of the two separates with the highest age values. Furthermore, the two applied K-Ar and Rb-Sr dating methods that are generally expected to provide similar age spreads in a case study such as this with a similar interpretation do not do so. A contribution by detrital components cannot yet be excluded, even if it is only theoretical, as such an explanation should provide similar K-Ar and Rb-Sr data, which is not the case.

Indeed, the Rb-Sr and K-Ar methods provide different results for the same separates, while they were expected to provide similar data for the impact of low-temperature events, such as those induced by burial or discrete thermal episodes. Basically, there is a need to combine them into a unique interpretation (e.g., discussion in [56]). If the higher K-Ar data relative to the Rb-Sr equivalents do not result from a detrital contribution, the glauconite-rich pellets had to undergo an event that modified and affected the Rb-Sr system more than the K-Ar system of the analyzed material, unless the K-Ar was not affected at all by a later, more discrete event. It cannot be excluded either that minerals trapped in the pellets, but external to the glauconite crystals, could have been modified or even added during later fluid episodes that affected the Rb-Sr system and not the K-Ar system of the pellets suspected to be somehow heterogeneous. Moreover, the older K-Ar values are close to the deposition time, which implies that the necessary event(s) that crystallized after deposition to explain the diverse mineralogy of the pellets with a combination glauconite-celadonite had to be of diagenetic origin with a limited impact on the earlier glauconite grains. This also implies a limited temperature increase between deposition and this diagenetic episode. In fact, except for the youngest K-Ar data that are close to the highest Rb-Sr ages, there is no direct correspondence amongst the K-Ar and the Rb-Sr data of strictly the same glauconite separates. Furthermore, the K-Ar data apply strictly to the K-bearing glauconite/celadonite carriers, while the Rb-Sr system may also be dependent on mineral phases bearing only or mostly Sr and negligible amounts or no Rb, such as carbonates and oxides found in the X-rayed pellets. Such components may be present in the pellets, even after leaching with dilute HCl, as highlighted by a recent study on similar materials [19].

### 5.2. Comparison of the K-Ar and Rb-Sr Data

The K-Ar data generated here and the previously released Rb-Sr results on the same glauconite-rich separates show a clear discrepancy that can be summarized by higher, older K-Ar age data at about 440, 415 and 390 Ma, while the Rb-Sr data are at about 405 Ma for the highest, i.e., oldest, and about 370 Ma for the lowest, i.e., youngest. An analytical problem may be discounted because the age differences are due to the used methods and not to the individual analytical data. Either one or several mineral phases of the pellets crystallized later than the glauconite/celadonite or their Rb-Sr system was altered more than the K-Ar system that exclusively concerns the K-carriers, during an event younger

than the deposition time. Such a late alteration cannot concern celadonite, a K-carrier that therefore yields "older" K-Ar data. Stein and Kish [30] also do not see evidence for a relationship between the widespread dolomitization and the ore deposit in the Missouri region. However, the occurrence of at least two regional Devonian disturbances suggests a Devonian (380–400 Ma) dolomitization and a Late Devonian-Early Mississippi (360–370 Ma) ore concentration. The late associated fluid movements could have been recorded by the Rb-Sr system of Sr-rich minerals trapped in the pellets and not by the K-carrier glauconite-celadonite minerals. These fluids also appear not to have impacted the K-Ar data apparently induced by earlier regional tectonic activity but to relate to the crystallization of the metal deposits. The oldest K-Ar ages of the glaucony are quite close and could correspond to an initial mineralizing episode soon after the deposition time of the glauconitic pellets. The fact that the K-Ar system seems not affected by possible repetitive low-temperature fluid flows is consistent with the celadonite not having crystallized after glauconite, unless the K-Ar system of the earlier glauconite was reset during a thermal episode that favored celadonite crystallization.

*5.3. Evaluation of the Previously Published Rb-Sr Data*

The Rb-Sr data of Posey et al. [20] provide age values for each glauconite separate based on a 0.7091 assumption for the initial $^{87}$Sr/$^{86}$Sr ratio. The obtained ages range from about 354 to 403 Ma with an average at 383 Ma (Table 5). In fact, this calculation raises two concerns: (1) the initial $^{87}$Sr/$^{86}$Sr ratio with a value close to the present-day marine value [57] is too high for marine glauconites of late Cambrian age, and (2) the average age suggests that Posey et al. [20] assume an event after deposition, which would justify the used high initial $^{87}$Sr/$^{86}$Sr ratio, but then the obtained age no longer relates to a stratigraphic crystallization but to a diagenetic alteration. In fact, the data are scattered above the isochron obtained by Kish and Stein [21] on separates of similar sedimentary horizons sampled in the nearby Magmont mine. However, the whole set of the individual Rb-Sr data, re-evaluated here, is scattered along an array giving an age of 404 ± 16 Ma with an unrealistic initial $^{87}$Sr/$^{86}$Sr ratio of 0.648 ± 0.054 (2σ) and a far too high MSWD at 13. The array is slightly steeper above the isochron of the samples from Magmont mine and slightly more dispersed. Both suggest the occurrence of detrital components in the analyzed separates (Figure 3). Clearly the K-Ar data are also scattered with a plausible contribution of either detrital crystals or a variable impact of a post-depositional discrete event that more or less altered some of the K-carrier minerals from analyzed separates. In performing the classical age calculation with Ludwig's [58] program for the Rb-Sr data published by Posey et al. [20], 6 of the Lower Bonneterre glauconite extracts plot along a theoretical isochron and give an age of 384 ± 11 (2σ) Ma for an initial $^{87}$Sr/$^{86}$Sr ratio of 0.710 ± 0.034 (2σ) with a MSWD of 1.18. A total of 4 of the 6 glaucony separates from Middle Bonneterre yield data plots sub-parallel to the initial Magmont isochron with an age of 368 ± 14 (2σ) Ma for an initial $^{87}$Sr/$^{86}$Sr ratio of 0.714 ± 0.028 (2σ) and a MSWD of 2.8. The data plots of 4 more glauconite separates of the Upper Bonneterre fit a 3rd array with an age of 369 ± 15 Ma (2σ) but with a far higher initial $^{87}$Sr/$^{86}$Sr ratio of 0.752 ± 0.058 (2σ) and a reasonable MSWD of 0.26. However, in an age of high technology and ultra-precise mass spectrometer analytical potential, it is important to remember that this reasoning is based on data determined about 40 years ago.

**Table 5.** The previously published Rb-Sr data of the glauconite-rich pellets studied here (data from Posey et al., 1983).

| Sample IDs | Sample Numbers | Rb (µg/g) | Sr (µg/g) | $^{87}Rbr/^{86}Sr$ | $^{87}Sr/^{86}Sr$ | Age (Ma) (*) |
|---|---|---|---|---|---|---|
| HC-1 | 1 | 258 | 3.75 | 222.6 | 1.9327 | 386 |
| 64W-133 | 2 | 254 | 3.08 | 273.4 | 2.1967 | 382 |
| | 3 | 271 | 2.58 | 361.4 | 2.6489 | 377 |
| | 4 | 251 | 5.44 | 143.4 | 1.4666 | 371 |
| | 5 | 253 | 3.77 | 216.4 | 1.8429 | 368 |
| 65W-32 | 6 | 253 | 3.75 | 218.3 | 1.8980 | 382 |
| 62W-141 | 7 | 246 | 3.32 | 242.1 | 2.0263 | 382 |
| CO-10 | 8 | 269 | 3.90 | 224.1 | 1.9601 | 392 |
| | 9 | 287 | 5.51 | 162.9 | 1.5307 | 354 |
| GT-1 | 10 | 243 | 3.99 | 195.0 | 1.8007 | 393 |
| CO-7 | 11 | 251 | 4.21 | 190.6 | 1.7584 | 387 |
| GT-5A | 12 | 242 | 4.50 | 170.0 | 1.6354 | 383 |
| | 13 | 243 | 5.53 | 136.7 | 1.4610 | 386 |
| 62W-126 | 14 | 262 | 2.29 | 407.2 | 3.0481 | 403 |
| | 15 | 252 | 3.99 | 202.8 | 1.8065 | 380 |
| 63W-89 | 16 | 254 | 3.72 | 221.8 | 1.9653 | 398 |

IDs stands for identities and (*) for age calculations with an initial $^{87}Sr/^{86}Sr$ ratio of 0.7091.

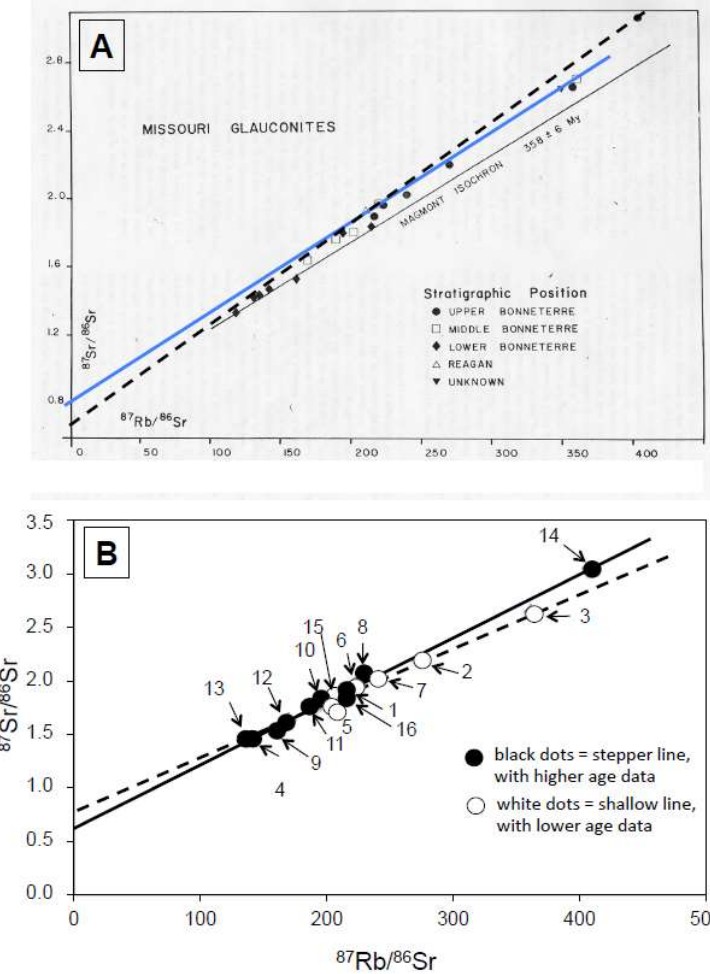

**Figure 3.** (**A**) A Rb-Sr display of the data from glauconite-rich pellets analyzed by Posey et al. [20] split into two lines (in black dots and full blue) of the samples studied here. The Magmont isochron published by Stein and Kish [30] is also given. (**B**) A view of the earlier Rb-Sr data from the samples studied here by K-Ar.

In summary, the Rb-Sr data of Posey et al. [20] confirm a scatter at least in the Middle and Upper Bonneterre sediments that was very slightly affected by a thermal increase that either accompanied or not metal-rich fluid flows at a fairly low burial depth and temperature. Obviously, this event affected accessory minerals hosted by the pellets or even favored new crystallizations of minerals enriched in Sr with an $^{87}$Sr/$^{86}$Sr ratio above that of the main glauconitic minerals of the pellets. Such an occurrence may have increased the overall $^{87}$Sr/$^{86}$Sr ratio of the heterogeneous pellets and consequently would have decreased the calculated ages. However, detritus can remain after sample purification as indicated by the XRD data and the fact that two Rb-Sr data are abnormally high in Harper's [55] diagram. Therefore, the Rb-Sr ages of potential stratigraphic value set the references at 384 ± 11 Ma (2σ) and 369 ± 15 Ma, the values overlapping analytically. The high initial $^{87}$Sr/$^{86}$Sr ratio of 0.710 used by Posey et al. [20] is justified due to an identified diagenetic impact that affected the host formations and, therefore, the pellets after the deposition of the host sediments, during one or even repetitive episodes at or between about 385 and 370 Ma ago.

### 5.4. How Do the K-Ar and the Earlier Published Rb-Sr Data of the Glauconites Combine?

The glauconite separates of most samples analyzed here were apparently not contaminated by detrital K-rich crystals, as both dating methods should have been affected and would display higher and more scattered age values than those obtained. The analytical dispersion seems, therefore, to have resulted from post-deposition diagenetic fluids that affected the Rb-Sr system more than, if at all, the K-Ar system by the plausible addition/subtraction of one or several Sr-rich and Rb-poor, and therefore K-poor mineral phases. A low-temperature episode, even at a low 60 °C is required, as the K-rich glauconitic material was apparently only slightly impacted during this diagenetic event.

All these prerequisites, together with high contents of metals in some of the studied glaucony pellets suggest that the studied material could have been more or less impregnated by flowing metal-rich fluids at a moderate temperature in an environment that was not touched by a tectonic–thermal event. The fact that the Rb-Sr system was the only one dating method visibly impacted, is probably linked with the specific behavior of Sr-rich and Rb-poor minerals trapped in the pellets that were apparently more sensitive to diagenetic interactions at low temperature than the glauconite/celadonite material. In summary, the results suggest that the Bonneterre Formation was a sort of regional drain for metal-rich fluids that may have percolated at a depth above 2000 m depth

Regional fluid flows interacting with soluble minerals such as carbonates, sulfates and/or oxides trapped in the analyzed pellets appear to have readjusted the Rb-Sr system of mineral phases external to the glauconite/celadonite crystals more than the K-Ar system of these later minerals. In summary, the generated data do not favor the occurrence of a significant thermal event of a regional extent but rather one to several metal-rich fluid circulations at low temperature in the glauconite-rich Bonneterre Formation. The pellets contain syn-depositional glaucony grains that the K-Ar system re-homogenized earlier during post-depositional diagenetic activities between 440 and 390 Ma that could have included celadonite crystallization. Metallic activity during this period cannot be ruled out. Later, between 385 and 370, low-temperature fluid flows probably dispersed various metals in the subsurface host-rocks that either affected or even favored the crystallization of calcite, sulfate and oxide-type minerals within the volume of the glauconite-rich pellets.

### 6. Conclusions

Glauconite-rich pellets of Late Paleozoic sediments from the state of Missouri that host metal concentrates were dated by the K-Ar method in order to complement previously published Rb-Sr ages. The preparation and purification steps of glauconite pellets are especially critical as there is a need for thorough purification to insure that not only the detrital counterparts are removed but also all Sr-rich accessory minerals such as the carbonates,

sulfates and oxides. Following this conclusion, the use of supplementary cleaning steps such as ultra-sonic shaking and/or leaching with dilute acids is strongly advocated.

Compared with the previously released Rb-Sr results, the K-Ar data obtained on the same separates outline clear age discrepancies that can be summarized as higher K-Ar age data at about 440, 415 and 390 Ma and lower Rb-Sr data at about 405 and 370 Ma. Most glaucony pellets were not significantly contaminated by detrital K-rich crystals, as both dating methods should have been affected in this case with variably increasing age data for both methods, even beyond deposition time. The visible analytical dispersion seems to have resulted from one or several successive diagenetic events that affected the Rb-Sr system more than the K-Ar system by a plausible addition/subtraction of one or several Sr-rich and Rb-poor and therefore K-poor phase(s). The studied pellets may have been impregnated by multi-episodic flows of metal-rich fluids at moderate temperatures in an environment not affected by a regional high thermal impact. The results suggest that the Bonneterre Formation was a regional drain for metal-rich fluids that percolated at a burial depth above 2000 m at least during periods at 410 + 5 Ma and 380 + 10 Ma. The further K-Ar age at about 440 Ma could have resulted from an early, more local post-depositional diagenetic that could have favored a separate crystallization of celadonite.

**Author Contributions:** Conceptualization: N.C.; methodology: N.C.; software: I.T.U.; analyses: A.A.; validation: N.C., I.T.U. and A.A.; writing original draft: N.C.; writing review: N.C.; All authors have read and agreed to the published version of the manuscript.

**Funding:** This research received no external funding.

**Institutional Review Board Statement:** Not applicable.

**Informed Consent Statement:** Not applicable.

**Data Availability Statement:** Data provided upon request.

**Acknowledgments:** Special cordial thanks are due to H.H. Posey of the University of North Carolina at the time he provided the glauconite separates studied here. We tried to contact him recently with a proposal to co-author this study but did not receive a reply. The elemental analyses and the K-Ar data were obtained at the earlier Centre de Géochimie de la Surface of the University of Strasbourg. D. Tisserand, R. Wendling and R. Winkler are sincerely thanked for their help in the present study and over the years. Also to be mentioned is the fact that this study did not benefit from any specific funding. Our sincere thanks are also for the two anonymous reviewers and their very constructive comments. One of them deserves especial thanks for having also helped improve the English presentation of the text.

**Conflicts of Interest:** The authors declare no conflict of interest.

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
