# Peer review of "Elemental and K-Ar Isotopic Signatures of Glauconite/Celadonite Pellets from a Metallic Deposit of Missouri: Genetic Implications for the Local Deposits"

_geosciences, doi:10.3390/geosciences12100387_

Round 1
Reviewer 1 Report
The arguments are robustly made, and the interpretation of data thorough.

Author Response
First of all we would like to sincerely thank the reviewer #1 for the time he did spend to correct and improve the text.
I have also tried to reply to his questions along the text:
1- The ages of any mixture containing detrital crystals is increasing when part of the depositional or authigenic components are dissolved.
2- Has been addressed by a slight change in the writing.
3- Any granular structure favors the crystallization of “foreign’ minerals in the wholes between the grains. This structure typical for glauconites has not be considered for fong, but has probably some importance for the addition of other minerals after deposition.
4- was addressed
5- bad term that has been changed
6- they did but the order of the authors changed.
7- I did not write it clearly but I believe so! No need to start a battle on that!
8- You are right, magnetic separation was never totally reliable for us! Depending on the composition on the grains, they deviated or not!
9- yes!
10- determining on the isotopic age!
11- yes in table 1
12- comment for the typesetter
13- correct, was already in the text
14- correct reading! I changed slightly the sentence
15- OK
16- comment for the typesetter; I relayed it in the cover letter
17- text modified
18- checked
19- The whole k content id for the whole grains not build exclusively by glaucony (-ite), therefore its decrease may be due by the “impurities” in the grains
20- I scratched it!
21- tried to improve!
22- did!
23- tried!
24- The K-Ar system was re-homogenized after deposition as suggested by the K-Ar data relative to the Rb-Sr ones
25- If their would only be additional detritals, the ages by both metodes should have remained in similar values.
26- yes

Reviewer 2 Report
The subject, main content and conclusions of the article are well defined in the author's abstract. The authors obtained and submitted for publication new K-Ar data for glauconite-celadonite globules from Cambrian sedimentary rocks hosting the Viburnum metal-bearing deposit in Missouri. The interpretation of these data from the point of view of the relationship between the obtained ages and mineralization is difficult for the following reasons:
1. The minerals of the vast majority of ore deposits are not geochronometers, i.e. they do not contain radioactive and radiogenic isotopes, which allow us to directly determine the age of the deposit. Therefore, when dating ore deposits, an additional problem is to establish the relationship between the time of formation of the dated mineral and the formation of ore deposits. On the basis of numerous studies cited in this article, the age of formation of "green globules" is usually considered early diagenesis of host sedimentary rocks, while the time of formation of ore bodies is obviously associated with the subsequent movement of fluids within these deposits.
2. By itself, K-Ar and Rb-Sr dating of "green globules" in Paleozoic and Precambrian sedimentary formations, even without ore bodies, but affected by secondary transformations, is not an easy task.
3. The dated globules contain two different minerals, both belonging to the 2:1 family of phyllosilicates, glauconite and celadonite. These minerals carry a significant amount of radioactive 40K and influence the calculated age, while objective independent information about simultaneous or sequential formation of these two mineral varieties is absent.
4. When the authors trying to interpret new K-Ar dating they are forced to compare them with the data obtained for the same mineral fractions by the Rb-Sr method four decades ago. The evolution of experimental methods of isotope dating that has taken place during this time raises additional and very important problems resulting from differences in laboratory sample preparation.
The observed systematic difference in K-Ar and Rb-Sr dating for the same mineral fractions (separates) of glauconite, when the K-Ar age is older than Rb-Sr age, is an extremely rare. The authors of the reviewed article, who are world-class experts in isotope geochemistry and sedimentary geochronology, encountered with such case and their approach to the interpretation of these differences, in my opinion, was quite professional.
Considerable attention in the article is properly paid to the preparation and purification of the selected fractions for geochronological study. This is very important, since many difficulties in the interpretation of isotope-geochronological data on glauconites that exist in the literature take place due to differences in the preparation of the analyzed fractions.
The authors studied the compositions of trace elements in glauconite-celadonite globules and showed that some of them contain metals from ore deposits, i.e. were affected by fluids. An X-ray study of the globules was carried out, as well as their ultrasonic cleaning with leaching in dilute acetic acid. As a result, the authors performed the traditional set of studies that exist in the literature for dating globular phyllosilicates. These studies, however, turned out to be sufficient only to conclude in the rank of a hypothesis that the observed difference in the K-Ar and Rb-Sr dates of the isolated mineral fractions is the result of a diagenetic event that affected the Rb-Sr system more than the K-Ar system. This is the result of the probable addition/removal of one or more phases rich in Sr and poor in Rb, and therefore also poor in K. Such a hypothesis looks like a logical conclusion of the study.
The article will certainly be of interest to readers involved in isotope dating of sedimentary rocks.
In my opinion, the peer-reviewed article deserves acceptance after minor editorial amendments related to the negligence of its design. For example, some of them: lines 262-263: "The obtained K-Ar ages range quite widely from 389.9±8.3 Ma to 460.0±9.9 Ma (Table 4)", while the table shows a higher value - 468.3±16.4 Ma; lines 298-299: "….their K-Ar ages were determined again (Table 3)", while the K-Ar ages are given in Table 4.
However, at the present time there is also an approach to the 2:1 dating of layered silicates, which is fundamentally different from the one chosen by the authors, and has been published so far only in Russian scientific journals. The approach includes a complex mineralogical-crystal-chemical study of these minerals, including methods such as Mössbauer and IR-spectroscopy, XRD, and in many cases makes it possible to clearly distinguish stratigraphically significant K-Ar and Rb-Sr isotope dates of glauconites corresponding to the time of early diagenesis from "rejuvenated" values ​​(For example, Zaitseva T.S. et al. The Lower Boundary of the Vendian in the Southern Urals as Evidenced by the Rb-Sr Age of Glauconites of the Bakeevo Formation // Stratigraphy and Geological Correlation. 2019. Vol. 27, No. 5. pp. 573–587, DOI: 10.1134/S0869593819050083). If the authors are interested in testing the mentioned approach on their own material, then they could contact Dr. T.S. Zaitseva, whose e-mail is attached: z-t-s@mail.ru.
Author Response
I would not say that the chemical approach was “only sufficient” to state that their was a supplementary diagenesis stage after deposition. This diagenetic event allowed to conclude that it is responsible for the metal deposits and not the orogenic episode!
I also disagree with “probable”! It is in fact the only possibility to explain the overall situation, which leads to a completely different interpretation of the MVT from this region! addition/removal of one or more phases rich in Sr and poor in Rb, and therefore also poor in K. Such a hypothesis looks like a logical conclusion of the study. (Thank you for agreeing)
Has been corrected in the text.
Unfortunately, the authors of this approach seem to have only published in their native language and I am totally ignorant in the Russian language. Apparently this article produces only Rb-Sr data probably obtained in Igor Gorokhov’s laboratory who I know from a long time ago. This publication is probably very informative but it does not allow K-Ar and Rb-Sr comparisons, which is he hart of our study.
I am no longer able to produce data as I am in an emeritus position and as the laboratory for K-Ar dating has been closed since my retirement! I thank the reviewer for this information.

Reviewer 3 Report
This is a good manuscript, which presents, and correctly interprets, the apparent discrepancies between the results of two isotopic methods for dating glauconite pellets. In my opinion, the presented model is coherent both from the mineralogical and geological points of view. Perhaps, alternative interpretations are possible, however the presented one is well supported by the offered data and the clear reasoning in the discussion. The organization, style, writing, length and presentation are adequate to the problem.
Some minor suggestions are commented below:
- The distinction between glauconite and celadonite in X ray diffraction is not straightforward, as the most of their peaks are coincident (actually the two are micas, hence they share basically the same structure). I assume that the performed distinction has been correct, as the usually employed criteria are specified in the cited literature. Nevertheless, it would be convenient to give some minimum information about it in methods. The most of the geological readers of the journal are not going to be conscious about the problem, but some clay-mineral expert readers could raise the question.
Formal aspects
- In tables 2, 3 and 4, you should specify the meaning of “(us)”. It is less evident than “IDs” and it is important for the comprehension of the text. In fact, I had problems to understand the commented differences in results between ultrasonic-treated samples and not, until I understood the meaning of the abbreviation.
- Figure 2. It would facilitate the understanding of the figure to specify in the caption that the different kinds of dots are simply collections of data that adjust to the different lines.
- Figure 2 caption. I suppose you mean Harper (1970), if not, the reference is lacking.
- Line 299 – (Table 4).
- Line 319 – (Table 4).
- Line 380 – (Table 5).
- Line 451 – If the sentence after “which” is explicative, as it seams, should be separated by a comma. I also ask myself (my mother language is not English) if it would be more correct “whose” instead of “which”.
Author Response
This review is identical to review #2 and therefore no supplementary reply has been made!